# Hard Negative Mixing for Contrastive Learning

**Yannis Kalantidis**  **Mert Bulent Sariyildiz**  **Noe Pion**

**Philippe Weinzaepfel**  **Diane Larlus**

NAVER LABS Europe
Grenoble, France

## Abstract

Contrastive learning has become a key component of self-supervised learning approaches for computer vision. By learning to embed two augmented versions of the same image close to each other and to push the embeddings of different images apart, one can train highly transferable visual representations. As revealed by recent studies, heavy data augmentation and large sets of negatives are both crucial in learning such representations. At the same time, data mixing strategies, either at the image or the feature level, improve both supervised and semi-supervised learning by synthesizing novel examples, forcing networks to learn more robust features. In this paper, we argue that an important aspect of contrastive learning, *i.e.* the effect of *hard negatives*, has so far been neglected. To get more meaningful negative samples, current top contrastive self-supervised learning approaches either substantially increase the batch sizes, or keep very large memory banks; increasing memory requirements, however, leads to diminishing returns in terms of performance. We therefore start by delving deeper into a top-performing framework and show evidence that harder negatives are needed to facilitate better and faster learning. Based on these observations, and motivated by the success of data mixing, we propose *hard negative mixing* strategies at the feature level, that can be computed on-the-fly with a minimal computational overhead. We exhaustively ablate our approach on linear classification, object detection, and instance segmentation and show that employing our hard negative mixing procedure improves the quality of visual representations learned by a state-of-the-art self-supervised learning method.

Project page: https://europe.naverlabs.com/mochi

## 1 Introduction

Contrastive learning was recently shown to be a highly effective way of learning visual representations in a self-supervised manner [8, 21]. Pushing the embeddings of two transformed versions of the same image (forming the positive pair) close to each other and further apart from the embedding of any other image (negatives) using a contrastive loss, leads to powerful and transferable representations. A number of recent studies [10, 17, 39] show that carefully hand-crafting the set of data augmentations applied to images is instrumental in learning such represen-

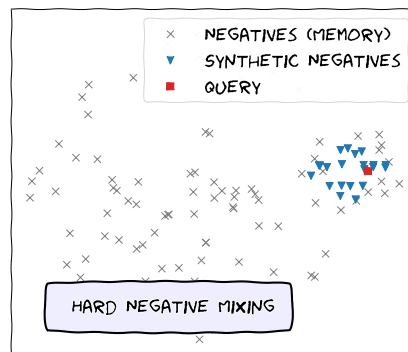

Figure 1: **MoCHi** generates synthetic hard negatives for each positive (query).

tations. We suspect that the right set of transformations provides more *diverse*, *i.e.* more challenging, copies of the same image to the model and makes the self-supervised (*proxy*) task harder. At the same time, data mixing techniques operating at either the pixel [41, 49, 50] or the feature level [40] help models learn more robust features that improve both supervised and semi-supervised learning on subsequent (*target*) tasks.

In most recent contrastive self-supervised learning approaches, the negative samples come from either the current batch or a memory bank. Because the number of negatives directly affects the contrastive loss, current top contrastive approaches either substantially increase the batch size [8], or keep large memory banks. Approaches like [31, 46] use memories that contain the whole training set, while the recent Momentum Contrast (or MoCo) approach of He et al. [21] keeps a queue with features of the last few batches as memory. The MoCo approach with the modifications presented in [10] (named MoCo-v2) currently holds the state-of-the-art performance on a number of target tasks used to evaluate the quality of visual representations learned in an unsupervised way. It is however shown [8, 21] that increasing the memory/batch size leads to diminishing returns in terms of performance: more negative samples does not necessarily mean *hard* negative samples.

In this paper, we argue that an important aspect of contrastive learning, *i.e.* the effect of hard negatives, has so far been neglected in the context of self-supervised representation learning. We delve deeper into learning with a momentum encoder [21] and show evidence that harder negatives are required to facilitate better and faster learning. Based on these observations, and motivated by the success of data mixing approaches, we propose *hard negative mixing*, *i.e.* feature-level mixing for hard negative samples, that can be computed on-the-fly with a minimal computational overhead. We refer to the proposed approach as **MoCHi**, that stands for "(**M**)ixing (**o**)f (**C**)ontrastive (**H**)ard negat(**i**)ves".

A toy example of the proposed hard negative mixing strategy is presented in Figure 1; it shows a t-SNE [29] plot after running MoCHi on 32-dimensional random embeddings on the unit hypersphere. We see that for each positive query (red square), the memory (gray marks) contains many easy negatives and few hard ones, *i.e.* many of the negatives are too far to contribute to the contrastive loss. We propose to mix only the hardest negatives (based on their similarity to the query) and synthesize new, hopefully also hard but more diverse, negative points (blue triangles).

**Contributions.** **a)** We delve deeper into a top-performing contrastive self-supervised learning method [21] and observe the need for harder negatives; **b)** We propose hard negative mixing, *i.e.* to synthesize hard negatives directly in the embedding space, on-the-fly, and adapted to each positive query. We propose to both mix pairs of the hardest existing negatives, as well as mixing the hardest negatives *with* the query itself; **c)** We exhaustively ablate our approach and show that employing hard negative mixing improves both the generalization of the visual representations learned (measured via their transfer learning performance), as well as the utilization of the embedding space, for a wide range of hyperparameters; **d)** We report competitive results for linear classification, object detection and instance segmentation, and further show that our gains over a state-of-the-art method are higher when pre-training for fewer epochs, *i.e.* MoCHi learns transferable representations faster.

## 2 Related work

Most early self-supervised learning methods are based on devising proxy classification tasks that try to predict the properties of a transformation (*e.g.* rotations, orderings, relative positions or channels) applied on a single image [12, 13, 16, 26, 32]. Instance discrimination [46] and CPC [33] were among the first papers to use contrastive losses for self-supervised learning. The last few months have witnessed a surge of successful approaches that also use contrastive learning losses. These include MoCo [10, 21], SimCLR [8, 9], PIRL [31], CMC [38] or SvAV [7]. In parallel, methods like [3, 5–7, 53, 27] build on the idea that clusters should be formed in the feature spaces, and use clustering losses together with contrastive learning or transformation prediction tasks.

Most of the top-performing contrastive methods leverage data augmentations [8, 10, 18, 21, 31, 38]. As revealed by recent studies [2, 17, 39, 43], heavy data augmentations applied to the same image are crucial in learning useful representations, as they modulate the hardness of the self-supervised task via the *positive pair*. Our proposed hard negative mixing technique, on the other hand, is changing the hardness of the proxy task from the side of the *negatives*.

A few recent works discuss issues around the selection of negatives in contrastive self-supervised learning [4, 11, 23, 45, 47, 22]. Iscen *et al.* [23] mine hard negatives from a large set by focusing on the features that are neighbors with respect to the Euclidean distance, but not when using a manifold distance defined over the nearest neighbor graph. Interested in approximating the underlying "true" distribution of negative examples, Chuang *et al.* [11] present a *debiased* version of the contrastive loss, in an effort to mediate the effect of false negatives. Wu *et al.* [45] present a variational extension to the InfoNCE objective that is further coupled with modified strategies for negative sampling, *e.g.* restricting negative sampling to a region around the query. In concurrent works, Cao *et al.* [4] propose a weight update correction for negative samples to decrease GPU memory consumption caused by weight decay regularization, while in [22] the authors propose a new algorithm that generates more challenging positive and hard negative pairs, on-the-fly, by leveraging adversarial examples.

**Mixing for contrastive learning.** Mixup [50] and its numerous variants [36, 40, 42, 49] have been shown to be highly effective data augmentation strategies when paired with a cross-entropy loss for supervised and semi-supervised learning. Manifold mixup [40] is a feature-space regularizer that encourages networks to be less confident for interpolations of hidden states. The benefits of interpolating have only recently been explored for losses other than cross-entropy [25, 36, 52]. In [36], the authors propose using mixup in the image/pixel space for self-supervised learning; in contrast, we create query-specific synthetic points *on-the-fly* in the embedding space. This makes our method way more computationally efficient and able to show improved results at a smaller number of epochs. The Embedding Expansion [25] work explores interpolating between embeddings for supervised metric learning on fine-grained recognition tasks. The authors use uniform interpolation between two positive and negative points, create a set of synthetic points and then select the hardest pair as negative. In contrast, the proposed MoCHi has no need for class annotations, performs no selection for negatives and only samples a single random interpolation between multiple pairs. What is more, in this paper we go beyond mixing negatives and propose mixing the positive with negative features, to get even harder negatives, and achieve improved performance. Our work is also related to metric learning works that employ *generators* [14, 51]. Apart from not requiring labels, our method exploits the memory component and has no extra parameters or loss terms that need to be optimized.

## 3 Understanding hard negatives in unsupervised contrastive learning

### 3.1 Contrastive learning with memory

Let $f$ be an encoder, *i.e.* a CNN for visual representation learning, that transforms an input image $\mathbf{x}$ to an *embedding* (or feature) vector $\mathbf{z} = f(\mathbf{x}), \mathbf{z} \in \mathbb{R}^d$. Further let $Q$ be a "memory bank" of size $K$, *i.e.* a set of $K$ embeddings in $\mathbb{R}^d$. Let the *query* $\mathbf{q}$ and *key* $\mathbf{k}$ embeddings form the positive pair, which is contrasted with every feature $\mathbf{n}$ in the bank of negatives ($Q$) also called the *queue* in [21]. A popular and highly successful loss function for contrastive learning [8, 21, 38] is the following:

$$\mathcal{L}_{\mathbf{q},\mathbf{k},Q} = - \log \frac{\exp(\mathbf{q}^T \mathbf{k}/\tau)}{\exp(\mathbf{q}^T \mathbf{k}/\tau) + \sum_{\mathbf{n} \in Q} \exp(\mathbf{q}^T \mathbf{n}/\tau)}, \tag{1}$$

where $\tau$ is a temperature parameter and all embeddings are $\ell_2$-normalized. In a number of recent successful approaches [8, 21, 31, 39] the query and key are the embeddings of two augmentations of the same image. The memory bank $Q$ contains negatives for each positive pair, and may be defined as an "external" memory containing every other image in the dataset [31, 38, 46], a queue of the last batches [21], or simply be every other image in the current minibatch [8].

The log-likelihood function of Eq (1) is defined over the probability distribution created by applying a `softmax` function for each input/query $\mathbf{q}$. Let $p_{z_i}$ be the matching probability for the query and feature $\mathbf{z}_i \in Z = Q \cup \{\mathbf{k}\}$, then the gradient of the loss with respect to the query $\mathbf{q}$ is given by:

$$\frac{\partial \mathcal{L}_{\mathbf{q},\mathbf{k},Q}}{\partial \mathbf{q}} = -\frac{1}{\tau} \left( (1 - p_k) \cdot \mathbf{k} - \sum_{\mathbf{n} \in Q} p_n \cdot \mathbf{n} \right), \quad \text{where } p_{z_i} = \frac{\exp(\mathbf{q}^T \mathbf{z}_i/\tau)}{\sum_{j \in Z} \exp(\mathbf{q}^T \mathbf{z}_j/\tau)}, \tag{2}$$

and $p_k, p_n$ are the matching probability of the key and negative feature, *i.e.* for $\mathbf{z}_i = \mathbf{k}$ and for $\mathbf{z}_i = \mathbf{n}$, respectively. We see that the contributions of the positive and negative logits to the loss are identical to the ones for a $(K+1)$-way cross-entropy classification loss, where the logit for the key corresponds to the query's *latent class* [1] and all gradients are scaled by $1/\tau$.

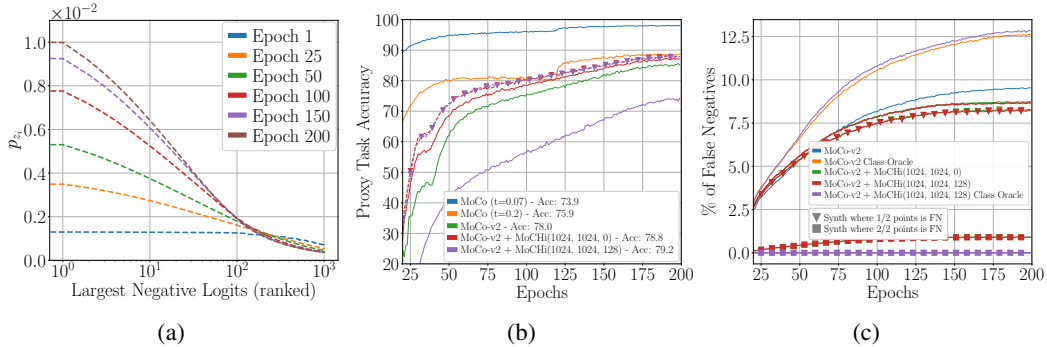

Figure 2: Training on ImageNet-100 dataset. **(a)** A histogram of the 1024 highest matching probabilities $p_{z_i}$, $z_i \in Q$ for MoCo-v2 [10], across epochs; logits are ranked by decreasing order and each line shows the average value of the matching probability over all queries; **(b)** Accuracy on *the proxy task*, *i.e.* percentage of queries where we rank the key over all negatives. Lines with triangle markers for MoCHi variants correspond to the proxy task accuracy after discarding the synthetic hard negatives. **(c)** Percentage of false negatives (FN), *i.e.* negatives from the same class as the query, among the highest 1024 (negative) logits. when using a class oracle. Lines with triangle (resp. square) markers correspond to the percentage of synthetic points for which one (resp. both) mixed points are FN. For Purple and orange lines the class oracle was used during training to discard FN.

## 3.2 Hard negatives in contrastive learning

Hard negatives are critical for contrastive learning [1, 19, 23, 30, 37, 44, 48]. Sampling negatives from the same batch leads to a need for larger batches [8] while sampling negatives from a memory bank that contains every other image in the dataset requires the time consuming task of keeping a large memory up-to-date [31, 46]. In the latter case, a trade-off exists between the "freshness" of the memory bank representations and the computational overhead for re-computing them as the encoder keeps changing. The Momentum Contrast (or MoCo) approach of He et al. [21] offers a compromise between the two negative sampling extremes: it keeps a queue of the latest $K$ features from the last batches, encoded with a second *key encoder* that trails the (main/query) encoder with a much higher momentum. For MoCo, the key feature **k** and all features in $Q$ are encoded with the key encoder.

**How hard are MoCo negatives?** In MoCo [21] (resp. SimCLR [8]) the authors show that increasing the memory (resp. batch) size, is crucial to getting better and harder negatives. In Figure 2a we visualize how hard the negatives are during training for MoCo-v2, by plotting the highest 1024 matching probabilities $p_i$ for ImageNet-100[1] and a queue of size $K = 16k$. We see that, although in the beginning of training (*i.e.* at epoch 0) the logits are relatively flat, as training progresses, fewer and fewer negatives offer significant contributions to the loss. This shows that most of the memory negatives are practically not helping a lot towards learning the proxy task.

**On the difficulty of the proxy task.** For MoCo [21], SimCLR [8], InfoMin [39], and other approaches that learn augmentation-invariant representations, we suspect the hardness of the proxy task to be directly correlated with the difficulty of the transformations set, *i.e.* hardness is modulated via the positive pair. We propose to experimentally verify this. In Figure 2b, we plot the *proxy task performance*, *i.e.* the percentage of queries where the key is ranked over all negatives, across training for MoCo [21] and MoCo-v2 [10]. MoCo-v2 enjoys a high performance gain over MoCo by three main changes: the addition of a Multilayer Perceptron (MLP) head, cosine learning rate schedule, and more challenging data augmentation. As we further discuss in the Appendix, only the latter of these three changes makes the proxy task harder to solve. Despite the drop in proxy task performance, however, further performance gains are observed for linear classification. In Section 4 we discuss how MoCHi gets a similar effect by modulating the proxy task through mixing harder negatives.

### 3.3 A class oracle-based analysis

In this section, we analyze the negatives for contrastive learning using a class oracle, *i.e.* the ImageNet class label annotations. Let us define *false negatives* (FN) as all negative features in the memory $Q$, that correspond to images of the same class as the query. Here we want to first quantify false negatives from contrastive learning and then explore how they affect linear classification performance. What is more, by using class annotations, we can train a contrastive self-supervised learning *oracle*, where we measure performance at the downstream task (linear classification) after disregarding FN from the negatives of each query during training. This has connections to the recent work of [24], where a contrastive loss is used in a supervised way to form positive pairs from images sharing the same label. Unlike [24], our oracle uses labels only for discarding negatives with the same label for each query, *i.e.* without any other change to the MoCo protocol.

In Figure 2c, we quantify the percentage of false negatives for the oracle run and MoCo-v2, when looking at highest 1024 negative logits across training epochs. We see that a) in all cases, as representations get better, more and more FNs (same-class logits) are ranked among the top; b) by discarding them from the negatives queue, the class oracle version (purple line) is able to bring same-class embeddings closer. Performance results using the class oracle, as well as a supervised upper bound trained with cross-entropy are shown in the bottom section of Figure 1. We see that the MoCo-v2 oracle recovers part of the performance relative to the supervised case, *i.e.* 78.0 (MoCo-v2, 200 epochs) $\rightarrow$ 81.8 (MoCo-v2 oracle, 200 epochs) $\rightarrow$ 86.2 (supervised).

## 4 Feature space mixing of hard negatives

In this section we present an approach for synthesizing hard negatives, *i.e.* by *mixing* some of the hardest negative features of the contrastive loss or the hardest negatives with the query. We refer to the proposed hard negative mixing approach as **MoCHi**, and use the naming convention MoCHi ($N$, $s$, $s'$), that indicates the three important hyperparameters of our approach, to be defined below.

### 4.1 Mixing the hardest negatives

Given a query $\mathbf{q}$, its key $\mathbf{k}$ and negative/queue features $\mathbf{n} \in Q$ from a queue of size $K$, the loss for the query is composed of logits $l(\mathbf{z}_i) = \mathbf{q}^T \mathbf{z}_i / \tau$ fed into a softmax function. Let $\tilde{Q} = \{\mathbf{n}_1, \ldots, \mathbf{n}_K\}$ be the *ordered* set of all negative features, such that: $l(\mathbf{n}_i) > l(\mathbf{n}_j), \forall i < j$, *i.e.* the set of negative features sorted by decreasing similarity to that particular query feature.

For each query, we propose to synthesize $s$ hard negative features, by creating convex linear combinations of pairs of its "hardest" existing negatives. We define the hardest negatives by truncating the ordered set $\tilde{Q}$, *i.e.* only keeping the first $N < K$ items. Formally, let $H = \{\mathbf{h}_1 \ldots, \mathbf{h}_s\}$ be the set of synthetic points to be generated. Then, a synthetic point $\mathbf{h}_k \in H$, would be given by:

$$\mathbf{h}_k = \frac{\tilde{\mathbf{h}}_k}{\|\tilde{\mathbf{h}}_k\|_2}, \text{ where } \tilde{\mathbf{h}}_k = \alpha_k \mathbf{n}_i + (1 - \alpha_k)\mathbf{n}_j, \tag{3}$$

$\mathbf{n}_i, \mathbf{n}_j \in \tilde{Q}^N$ are randomly chosen negative features from the set $\tilde{Q}^N = \{\mathbf{n}_1, \ldots, \mathbf{n}_N\}$ of the closest $N$ negatives, $\alpha_k \in (0, 1)$ is a randomly chosen mixing coefficient and $\|\cdot\|_2$ is the $\ell_2$-norm. After mixing, the logits $l(\mathbf{h}_k)$ are computed and appended as further *negative* logits for query $\mathbf{q}$. The process repeats for each query in the batch. Since all other logits $l(\mathbf{z}_i)$ are already computed, the extra computational cost only involves computing $s$ dot products between the query and the synthesized features, which would be computationally equivalent to increasing the memory by $s << K$.

### 4.2 Mixing for even harder negatives

As we are creating hard negatives by convex combinations of the existing negative features, and if one disregards the effects of the $\ell_2$-normalization for the sake of this analysis, the generated features will lie inside the convex hull of the hardest negatives. Early during training, where in most cases there is no linear separability of the query with the negatives, this synthesis may result in negatives much harder than the current. As training progresses, and assuming that linear separability is achieved, synthesizing features this way does not necessarily create negatives harder than the hardest ones present; it does however still *stretch* the space around the query, pushing the memory negatives further

and increasing the uniformity of the space (see Section 4.3). This space stretching effect around queries is also visible in the t-SNE projection of Figure 1.

To explore our intuition to the fullest, we further propose to mix the *query* with the hardest negatives to get even harder negatives for the proxy task. We therefore further synthesize $s'$ synthetic hard negative features for each query, by mixing its feature with a randomly chosen feature from the hardest negatives in set $\tilde{Q}^N$. Let $H' = \{\mathbf{h}'_1 \ldots, \mathbf{h}'_{s'}\}$ be the set of synthetic points to be generated by mixing the query and negatives, then, similar to Eq. (3), the synthetic points $\mathbf{h}'_k = \tilde{\mathbf{h}}'_k / \|\tilde{\mathbf{h}}'_k\|_2$, where $\tilde{\mathbf{h}}'_k = \beta_k \mathbf{q} + (1 - \beta_k)\mathbf{n}_j$, and $\mathbf{n}_j$ is a randomly chosen negative feature from $\tilde{Q}^N$, while $\beta_k \in (0, 0.5)$ is a randomly chosen mixing coefficient for the query. Note that $\beta_k < 0.5$ guarantees that the query's contribution is always smaller than the one of the negative. Same as for the synthetic features created in Section 4.1, the logits $l(\mathbf{h}'_k)$ are computed and added as further *negative* logits for query $\mathbf{q}$. Again, the extra computational cost only involves computing $s'$ dot products between the query and negatives. In total, the computational overhead of MoCHi is essentially equivalent to increasing the size of the queue/memory by $s + s' << K$.

### 4.3   Discussion and analysis of MoCHi

Recent approaches like [8, 10] use a Multi-layer Perceptron (MLP) head instead of a linear layer for the embeddings that participate in the contrastive loss. This means that the embeddings whose dot products contribute to the loss, are not the ones used for target tasks–a lower-layer embedding is used instead. Unless otherwise stated, we follow [8, 10] and use a 2-layer MLP head on top of the features we use for downstream tasks. We always mix hard negatives in the space of the loss.

**Is the proxy task more difficult?**   Figure 2b shows the *proxy task* performance for two variants of MoCHi, when the synthetic features are included (lines with no marker) and without (lines with triangle marker). We see that when mixing only pairs of negatives ($s' = 0$, green lines), the model does learn faster, but in the end the proxy task performance is similar to the baseline case. In fact, as features converge, we see that $\max l(\mathbf{h}_k) < \max l(\mathbf{n}_j), \mathbf{h}_k \in H, \mathbf{n}_j \in \tilde{Q}^N$. This is however not the case when synthesizing negatives by further mixing them *with* the query. As we see from Figure 2b, at the end of training, $\max l(\mathbf{h}'_k) > \max l(\mathbf{n}_j), \mathbf{h}'_k \in H'$, *i.e.* although the final performance for the proxy task when discarding the synthetic negatives is similar to the MoCo-v2 baseline (red line with triangle marker), when they are taken into account, the final performance is much lower (red line without markers). Through MoCHi we are able to modulate the hardness of the proxy task through the hardness of the negatives; in the next section we experimentally ablate that relationship.

**Oracle insights for MoCHi.**   From Figure 2c we see that the percentage of synthesized features obtained by mixing two false negatives (lines with square markers) increases over time, but remains very small, *i.e.* around only 1%. At the same time, we see that about 8% of the synthetic features are fractionally false negatives (lines with triangle markers), *i.e.* at least one of its two components is a false negative. For the oracle variants of MoCHi, we also do not allow false negatives to participate in synthesizing hard negatives. From Table 1 we see that not only the MoCHi oracle is able to get a higher upper bound (82.5 vs 81.8 for MoCo-v2), further closing the difference to the cross entropy upper bound, but we also show in the Appendix that, after longer training, the MoCHi oracle is able to recover *most* of the performance loss versus using cross-entropy, *i.e.* 79.0 (MoCHi, 200 epochs) $\rightarrow$ 82.5 (MoCHi oracle, 200 epochs) $\rightarrow$ 85.2 (MoCHi oracle, 800 epochs) $\rightarrow$ 86.2 (supervised).

It is noteworthy that by the end of training, MoCHi exhibits slightly lower percentage of false negatives in the top logits compared to MoCo-v2 (rightmost values of Figure 2c). This is an interesting result: MoCHi adds synthetic negative points that are (at least partially) false negatives and is pushing embeddings of the same class apart, but at the same time it exhibits higher performance for linear classification on ImageNet-100. That is, it seems that although the absolute similarities of same-class features may decrease, the method results in a more linearly separable space. This inspired us to further look into how having synthetic hard negatives impacts the *utilization* of the embedding space.

**Measuring the utilization of the embedding space.**   Very recently, Wang and Isola [43] presented two losses/metrics for assessing contrastive learning representations. The first measures the *alignment* of the representation on the hypersphere, *i.e.* the absolute distance between representations with the same label. The second measures the *uniformity* of their distribution on the hypersphere, through

measuring the logarithm of the average pairwise Gaussian potential between all embeddings. In Figure 3c, we plot these two values for a number of models, when using features from all images in the ImageNet-100 validation set. We see that MoCHi highly improves the uniformity of the representations compared to both MoCo-v2 and the supervised models. This further supports our hypothesis that MoCHi allows the proxy task to learn to better utilize the embedding space. In fact, we see that the supervised model leads to high alignment but very low uniformity, denoting features targeting the classification task. On the other hand, MoCo-v2 and MoCHi have much better spreading of the underlying embedding space, which we experimentally know leads to more *generalizable* representations, *i.e.* both MoCo-v2 and MoCHi outperform the supervised ImageNet-pretrained backbone for transfer learning (see Figure 3c).

## 5 Experiments

We learn representations on two datasets, the common ImageNet-1K [35], and its smaller ImageNet-100 subset, also used in [36, 38]. All runs of MoCHi are based on MoCo-v2. We developed our approach on top of the official public implementation of MoCo-v2[2] and reproduced it on our setup; other results are copied from the respective papers. We run all experiments on 4 GPU servers. For linear classification on ImageNet-100 (resp. ImageNet-1K), we follow the common protocol and report results on the validation set. We report performance after learning linear classifiers for 60 (resp. 100) epochs, with an initial learning rate of 10.0 (30.0), a batch size of 128 (resp. 512) and a step learning rate schedule that drops at epochs 30, 40 and 50 (resp. 60, 80). For training we use $K = 16$k (resp. $K = 65$k). For MoCHi, we also have a warm-up of 10 (resp. 15) epochs, *i.e.* for the first epochs we do not synthesize hard

| Method | Top1 % ($\pm\sigma$) | diff (%) |
|---|---|---|
| MoCo [21] | 73.4 | |
| MoCo + iMix [36] | 74.2[‡] | 0.8 |
| CMC [38] | 75.7 | |
| CMC + iMix [36] | 75.9[‡] | 0.2 |
| MoCo [21]* ($t = 0.07$) | 74.0 | |
| MoCo [21]* ($t = 0.2$) | 75.9 | |
| MoCo-v2 [10]* | 78.0 ($\pm$0.2) | |
|    + MoCHi (1024, 1024, 128) | **79.0** ($\pm$0.4) | **1.0** |
|    + MoCHi (1024, 256, 512) | **79.0** ($\pm$0.4) | **1.0** |
|    + MoCHi (1024, 128, 256) | **78.9** ($\pm$0.5) | **0.9** |
| *Using Class Oracle* | | |
|    MoCo-v2* | 81.8 | |
|     + MoCHi (1024, 1024, 128) | 82.5 | |
| *Supervised (Cross Entropy)* | 86.2 | |

Table 1: Results on ImageNet-100 after training for 200 epochs. The bottom section reports results when using a class oracle (see Section 3.3). * denotes reproduced results, [‡] denotes results visually extracted from Figure 4 in [36]. The parameters of MoCHi are $(N, s, s')$.

negatives. For ImageNet-1K, we report accuracy for a single-crop testing. For object detection on PASCAL VOC [15] we follow [21] and fine-tune a Faster R-CNN [34], R50-C4 on `trainval07+12` and test on `test2007`. We use the open-source `detectron2`[3] code and report the common AP, AP50 and AP75 metrics. Similar to [21], we do *not* perform hyperparameter tuning for the object detection task. See the Appendix for more implementation details.

**A note on reporting variance in results.** It is unfortunate that many recent self-supervised learning papers do not discuss variance ; in fact only papers from highly resourceful labs [21, 8, 39] report averaged results, but not always the variance. This is generally understandable, as *e.g.* training and evaluating a ResNet-50 model on ImageNet-1K using 4 V100 GPUs take about 6-7 days. In this paper, we tried to verify the variance of our approach for a) self-supervised pre-training on ImageNet-100, *i.e.* we measure the variance of MoCHi runs by training a model multiple times from scratch (Table 1), and b) the variance in the fine-tuning stage for PASCAL VOC and COCO (Tables 2, 3). It was unfortunately computationally infeasible for us to run multiple MoCHi pre-training runs for ImageNet-1K. In cases where standard deviation is presented, it is measured over at least 3 runs.

### 5.1 Ablations and results

We performed extensive ablations for the most important hyperparameters of MoCHi on ImageNet-100 and some are presented in Figures 3a and 3b, while more can be found in the Appendix. In general we see that multiple MoCHi configurations gave consistent gains over the MoCo-v2 baseline [10] for linear classification, with the top gains presented in Figure 1 (also averaged over 3 runs). We further show performance for different values of $N$ and $s$ in Figure 3a and a table of gains for $N = 1024$ in

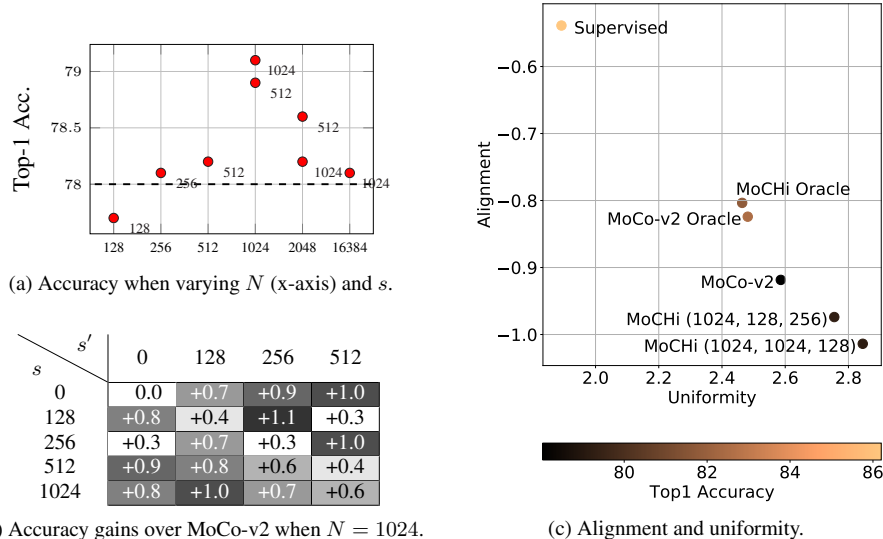

(a) Accuracy when varying $N$ (x-axis) and $s$.

| $s$ \ $s'$ | 0 | 128 | 256 | 512 |
|---|---|---|---|---|
| 0 | 0.0 | +0.7 | +0.9 | +1.0 |
| 128 | +0.8 | +0.4 | +1.1 | +0.3 |
| 256 | +0.3 | +0.7 | +0.3 | +1.0 |
| 512 | +0.9 | +0.8 | +0.6 | +0.4 |
| 1024 | +0.8 | +1.0 | +0.7 | +0.6 |

(b) Accuracy gains over MoCo-v2 when $N = 1024$.

(c) Alignment and uniformity.

Figure 3: Results on the validation set of ImageNet-100. **(a)** Top-1 Accuracy for different values of $N$ (x-axis) and $s$; the dashed black line is MoCo-v2. **(b)** Top-1 Accuracy gains (%) over MoCo-v2 (top-left cell) when $N = 1024$ and varying $s$ (rows) and $s'$ (columns). **(c)** Alignment and uniformity metrics [43]. The x/y axes correspond to $-\mathcal{L}_{uniform}$ and $-\mathcal{L}_{align}$, respectively.

Figure 3b; we see that a large number of MoCHi combinations give consistent performance gains. Note that the results in these two tables are *not averaged* over multiple runs (for MoCHi combinations where we had multiple runs, only the first run is presented for fairness). In other ablations (see Appendix), we see that MoCHi achieves gains (+0.7%) over MoCo-v2 also when training for 100 epochs. Table 1 presents comparisons between the best-performing MoCHi variants and reports gains over the MoCo-v2 baseline. We also compare against the published results from [36] a recent method that uses mixup in pixel space to synthesize harder images.

**Comparison with the state of the art on ImageNet-1K, PASCAL VOC and COCO.** In Table 2 we present results obtained after training on the ImageNet-1K dataset. Looking at the average negative logits plot and because both the queue and the dataset are about an order of magnitude larger for this training dataset we mostly experiment with smaller values for $N$ than in ImageNet-100. Our main observations are the following: a) MoCHi does not show performance gains over MoCo-v2 for linear classification on ImageNet-1K. We attribute this to the biases induced by training with hard negatives on the same dataset as the downstream task: Figures 3c and 2c show how hard negative mixing reduces alignment and increases uniformity for the dataset that is used during training. MoCHi still retains state-of-the-art performance. b) MoCHi helps the model learn faster and achieves high performance gains over MoCo-v2 for transfer learning after only 100 epochs of training. c) The harder negative strategy presented in Section 4.2 helps a lot for shorter training. d) In 200 epochs MoCHi can achieve *performance similar to MoCo-v2 after 800 epochs* on PASCAL VOC. e) From all the MoCHi runs reported in Table 2 as well as in the Appendix, we see that performance gains are consistent across multiple hyperparameter configurations.

In Table 3 we present results for object detection and semantic segmentation on the COCO dataset [28]. Following He et al. [21], we use Mask R-CNN [20] with a C4 backbone, with batch normalization tuned and synchronize across GPUs. The image scale is in [640, 800] pixels during training and is 800 at inference. We fine-tune all layers end-to-end on the `train2017` set (118k images) and evaluate on `val2017`. We adopt feature normalization as in [21] when fine-tuning. MoCHi and MoCo use the same hyper-parameters as the ImageNet supervised counterpart (*i.e.* we did not do any method-specific tuning). From Table 3 we see that MoCHi displays consistent gains over both the supervised baseline and MoCo-v2, for both 100 and 200 epoch pre-training. In fact, MoCHi is able to reach the AP performance similar to supervised pre-training for instance segmentation (33.2) after only 100 epochs of pre-training.

| Method | IN-1k Top1 | VOC 2007 AP$_{50}$ | AP | AP$_{75}$ |
|---|---|---|---|---|
| | | *100 epoch training* | | |
| MoCo-v2 [10]* | 63.6 | 80.8 (±0.2) | 53.7 (±0.2) | 59.1 (±0.3) |
| + MoCHi (256, 512, 0) | 63.9 | 81.1 (±0.1) (0.4) | 54.3 (±0.3) (0.7) | 60.2 (±0.1) (1.2) |
| + MoCHi (256, 512, 256) | 63.7 | **81.3** (±0.1) **(0.6)** | 54.6 (±0.3) (1.0) | 60.7 (±0.8) (1.7) |
| + MoCHi (128, 1024, 512) | 63.4 | 81.1 (±0.1) (0.4) | **54.7** (±0.3) **(1.1)** | **60.9** (±0.1) **(1.9)** |
| | | *200 epoch training* | | |
| SimCLR [8] (8k batch size, from [10]) | 66.6 | | | |
| MoCo + Image Mixture [36] | 60.8 | 76.4 | | |
| InstDis [46]$^{†}$ | 59.5 | 80.9 | 55.2 | 61.2 |
| MoCo [21] | 60.6 | 81.5 | 55.9 | 62.6 |
| PIRL [31]$^{†}$ | 61.7 | 81.0 | 55.5 | 61.3 |
| MoCo-v2 [10] | 67.7 | 82.4 | 57.0 | 63.6 |
| InfoMin Aug. [39] | 70.1 | 82.7 | **57.6** | **64.6** |
| MoCo-v2 [10]* | 67.9 | 82.5 (±0.2) | 56.8 (±0.1) | 63.3 (±0.4) |
| + MoCHi (1024, 512, 256) | 68.0 | 82.3 (±0.2) (0.2) | 56.7 (±0.2) (0.1) | 63.8 (±0.2) (0.5) |
| + MoCHi (512, 1024, 512) | 67.6 | 82.7 (±0.2) (0.2) | 57.1 (±0.1) (0.3) | 64.1 (±0.3) (0.8) |
| + MoCHi (256, 512, 0) | 67.7 | 82.8 (±0.2) (0.3) | 57.3 (±0.2) (0.5) | 64.1 (±0.1) (0.8) |
| + MoCHi (256, 512, 256) | 67.6 | 82.6 (±0.2) (0.1) | 57.2 (±0.3) (0.4) | 64.2 (±0.5) (0.9) |
| + MoCHi (256, 2048, 2048) | *67.0* | 82.5 (±0.1) ( 0.0) | 57.1 (±0.2) (0.3) | 64.4 (±0.2) (1.1) |
| + MoCHi (128, 1024, 512) | *66.9* | 82.7 (±0.2) (0.2) | 57.5 (±0.3) (0.7) | 64.4 (±0.4) (1.1) |
| | | *800 epoch training* | | |
| SvAV [7] | *75.3* | 82.6 | 56.1 | 62.7 |
| MoCo-v2 [10] | 71.1 | 82.5 | **57.4** | 64.0 |
| MoCo-v2[10]* | 69.0 | 82.7 (±0.1) | 56.8 (±0.2) | 63.9 (±0.7) |
| + MoCHi (128, 1024, 512) | 68.7 | **83.3** (±0.1) (0.6) | 57.3 (±0.2) (0.5) | **64.2** (±0.4) (0.3) |
| Supervised [21] | 76.1 | 81.3 | 53.5 | 58.8 |

Table 2: Results for linear classification on **ImageNet-1K** and object detection on **PASCAL VOC** with a ResNet-50 backbone. Wherever standard deviation is reported, it refers to multiple runs for the fine-tuning part. For MoCHi runs we also report in parenthesis the difference to MoCo-v2. * denotes reproduced results. $^{†}$ results are copied from [21]. We **bold** (resp. underline) the highest results overall (resp. for MoCHi).

| Pre-train | AP$^{bb}$ | AP$^{bb}_{50}$ | AP$^{bb}_{75}$ | AP$^{mk}$ | AP$^{mk}_{50}$ | AP$^{mk}_{75}$ |
|---|---|---|---|---|---|---|
| Supervised [21] | 38.2 | 58.2 | 41.6 | 33.3 | 54.7 | 35.2 |
| | | | *100 epoch pre-training* | | | |
| MoCo-v2 [10]* | 37.0 (±0.1) | 56.5 (±0.3) | 39.8 (±0.1) | 32.7 (±0.1) | 53.3 (±0.2) | 34.3 (±0.1) |
| + MoCHi (256, 512, 0) | 37.5 (±0.1) (0.5) | 57.0 (±0.1) (0.5) | 40.5 (±0.2) (0.7) | 33.0 (±0.1) (0.3) | 53.9 (±0.2) (0.6) | 34.9 (±0.1) (0.6) |
| + MoCHi (128, 1024, 512) | **37.8** (±0.1) **(0.8)** | **57.2** (±0.0) **(0.7)** | **40.8** (±0.2) **(1.0)** | **33.2** (±0.0) **(0.5)** | **54.0** (±0.2) **(0.7)** | **35.4** (±0.1) **(1.1)** |
| | | | *200 epoch pre-training* | | | |
| MoCo [21] | 38.5 | 58.3 | 41.6 | 33.6 | 54.8 | 35.6 |
| MoCo (1B image train) [21] | 39.1 | 58.7 | 42.2 | 34.1 | 55.4 | 36.4 |
| InfoMin Aug. [39] | 39.0 | 58.5 | 42.0 | 34.1 | 55.2 | 36.3 |
| MoCo-v2 [10]* | 39.0 (±0.1) | 58.6 (±0.1) | 41.9(±0.3) | 34.2 (±0.1) | 55.4 (±0.1) | 36.2 (±0.2) |
| + MoCHi (256, 512, 0) | 39.2 (±0.1) (0.2) | 58.8 (±0.1) (0.2) | 42.4 (±0.2) (0.5) | 34.4 (±0.1) (0.2) | 55.6 (±0.1) (0.2) | 36.7 (±0.1) (0.5) |
| + MoCHi (128, 1024, 512) | 39.2 (±0.1) (0.2) | 58.9 (±0.2) (0.3) | 42.4 (±0.3) (0.5) | 34.3 (±0.1) (0.2) | 55.5 (±0.1) (0.1) | 36.6 (±0.1) (0.4) |
| + MoCHi (512, 1024, 512) | **39.4** (±0.1) **(0.4)** | **59.0** (±0.1) **(0.4)** | **42.7** (±0.1) **(0.8)** | **34.5** (±0.0) **(0.3)** | **55.7** (±0.2) **(0.3)** | **36.7** (±0.1) **(0.5)** |

Table 3: Object detection and instance segmentation results on **COCO** with the ×1 training schedule and a C4 backbone. * denotes reproduced results.

# 6 Conclusions

In this paper we analyze a state-of-the-art method for self-supervised contrastive learning and identify the need for harder negatives. Based on that observation, we present a hard negative mixing approach that is able to improve the quality of representations learned in an unsupervised way, offering better transfer learning performance as well as a better utilization of the embedding space. What is more, we show that we are able to learn generalizable representations *faster*, something important considering the high compute cost of self-supervised learning. Although the hyperparameters needed to get maximum gains are specific to the training set, we find that multiple MoCHi configurations provide considerable gains, and that hard negative mixing consistently has a positive effect on transfer learning performance.

## Broader Impact

**Self-supervised tasks and dataset bias.** Prominent voices in the field advocate that self-supervised learning will play a central role during the next years in the field of AI. Not only representations learned using self-supervised objectives directly reflect the biases of the underlying dataset, but also it is the responsibility of the scientist to explicitly try to minimize such biases. Given that, the larger the datasets, the harder it is to properly investigate biases in the corpus, we believe that notions of *fairness* need to be explicitly tackled during the self-supervised *optimization*, *e.g.* by regulating fairness on protected attributes. This is especially important for systems whose decisions affect humans and/or their behaviours.

**Self-supervised learning, compute and impact on the environment.** On the one hand, self-supervised learning involves training large models on very large datasets, on long periods of time. As we also argue in the main paper, the computational cost of every self-supervised learning paper is very high: pre-training for 200 epochs on the relatively small ImageNet-1K dataset requires around 24 GPU days (6 days on 4 GPUs). In this paper we show that, by mining harder negatives, one can get higher performance after training for fewer epochs; we believe that it is indeed important to look deeper into self-supervised learning approaches that utilize the training dataset better and learn generalizable representations faster.

Looking at the bigger picture, however, we believe that research in self-supervised learning is highly justified in the long run, despite its high computational cost, for two main reasons. First, the goal of self-supervised learning is to produce models whose representations generalize better and are therefore potentially useful for many subsequent tasks. Hence, having strong models pre-trained by self-supervision would reduce the environmental impact of deploying to multiple new downstream tasks. Second, representations learned from huge corpora have been shown to improve results when directly fine-tuned, or even used as simple feature extractors, on smaller datasets. Most socially minded applications and tasks fall in this situation where they have to deal with limited annotated sets, because of a lack of funding, hence they would directly benefit from making such pretraining models available. Given the considerable budget required for large, high quality datasets, we foresee that strong generalizable representations will greatly benefit socially mindful tasks more than *e.g.* a multi-billion dollar industry application, where the funding to get large clean datasets already exists.

## Acknowledgements

This work is part of MIAI@Grenoble Alpes (ANR-19-P3IA-0003).

## Footnotes

[1]In this section we study contrastive learning for MoCo [21] on ImageNet-100, a subset of ImageNet consisting of 100 classes introduced in [38]. See Section 5 for details on the dataset and experimental protocol.

[2]https://github.com/facebookresearch/moco

[3]https://github.com/facebookresearch/detectron2

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
