[Supplementary Material]

# Hard Negative Mixing for Contrastive Learning

**Yannis Kalantidis**  **Mert Bulent Sariyildiz**  **Noe Pion**

**Philippe Weinzaepfel**  **Diane Larlus**

NAVER LABS Europe
Grenoble, France

# Appendices

## Appendix A    Details for the uniformity experiment

The uniformity experiment is based on Wang and Isola [53]. We follow the same definitions of the losses/metrics as presented in the paper. The alignment loss is given by:

$$\mathcal{L}_{\mathsf{align}}(f;\alpha) := - \mathop{\mathbb{E}}_{(\mathbf{x},\mathbf{y}) \sim p_{\mathsf{pos}}} \left[ \| f(\mathbf{x}) - f(\mathbf{y}) \|_2^\alpha \right], \quad \alpha > 0,$$

while the uniformity loss is given by:

$$\mathcal{L}_{\mathsf{uniform}}(f;t) := \log \mathop{\mathbb{E}}_{\mathbf{x},\mathbf{y} \overset{\mathsf{i.i.d.}}{\sim} p_{\mathsf{data}}} \left[ e^{-t\|f(\mathbf{x})-f(\mathbf{y})\|_2^2} \right], \quad t > 0,$$

where $\alpha, t$ are weighting parameters and f is the feature encoder (*i.e.* minus the MLP head for MoCo-v2 and MoCHi). We set $\alpha = 2$ and $t = 2$. All features were L2-normalized, as the metrics are defined on the hypersphere. $p_{\mathsf{pos}}$ denotes the joint distribution of pairs of positive samples, and $p_{\mathsf{data}}$ is the distribution of the data. Note that $p_{\mathsf{pos}}$ is task-specific; here we use the class oracle, *i.e.* the ImageNet-100 labels, to define the positive samples. We use the publicly available implementation supplied by the authors[1]; we modify the alignment implementation to reflect the fact that we obtain the positives based on the class oracle. In the Figure, we report the two metrics ($-\mathcal{L}_{uniform}$ and $-\mathcal{L}_{align}$) for models trained on ImageNet-100 using all embeddings of the validation set.

## Appendix B    Further analysis on hard negative mixing

### B.1    Proxy task: Effect of MLP and Stronger Augmentation

Following our discussion in Section 3, we wanted to verify that hardness of the proxy task for MoCo [19] is directly correlated to the difficulty of the transformations set, *i.e.* proxy task hardness can modulated via the positive pair. In Figure 1, we plot the *proxy task performance*, *i.e.* the percentage of queries where the key is ranked over all negatives, across training for MoCo [19], MoCo-v2 [10] and some variants inbetween. In Figure 1, we track the proxy task performance when progressively moving from MoCo to MoCo-v2, *i.e.* a) switching to a cosine learning rate schedule (gray line–no noticeable change in performance after 200 epochs); b) adding a Multilayer Perceptron head (cyan line–no noticeable change in performance after 200 epochs); c) adding a more

Figure 1: Proxy task performance over 200 epochs of training on ImageNet-100. For all methods we use the same $\tau = 0.2$.

challenging data augmentation (green line–a drop in proxy task performance). The latter is equivalent to MoCo-v2. For completeness we further show a MoCHi run with a large number of synthetic features (red line–large drop in in proxy task performance).

It is worth noting that the temperature $\tau$ of the softmax is a hyper-parameter that highly affects the rate of learning and therefore performance in the proxy task. As mentioned above, all results in Figure 1 are for the same $\tau = 0.2$.

## B.2 Hard negative mixing variants not discussed in the main text

While developing MoCHi, we considered a number of different mixing strategies in feature space. Many of those resulted in lower performance while others performed on par but were unnecessarily more complicated. We found the two strategies presented in Sections 4.1 and 4.2 of the main paper to be both the best performing and also complementary. Here, we briefly mention some other ideas that didn't make the cut.

**Mixing using keys instead of queries.** For MoCHi, the "top" negatives are defined via the negative logits, *i.e.* how far each memory negative is to a *query*. We also experimented when the ranking comes from a *key*, *i.e.* using the key to define the ordering of the set $\tilde{Q}$. We ablated this for both when mixing pairs of negative as well as when mixing the query with negatives. In the vast majority of the cases, results were on average about 0.2% lower, across multiple configurations. Note that this would also involve having to compute the dot products of the key with the memory, something that would induce further computational overhead.

**Mixing keys with negatives.** For MoCHi, in Section 4.2 we propose to synthesize $s'$ synthetic hard negative features for each query, by mixing its feature with a randomly chosen feature from the hardest negatives in set $\tilde{Q}^N$. We could also create such negatives by mixing the *key* the same way, *i.e.* the synthetic points created this way would be given by $\mathbf{h}''_k = \tilde{\mathbf{h}}''/\|\tilde{\mathbf{h}}''_k\|_2$, where $\tilde{\mathbf{h}}''_k = \beta_k \mathbf{k} + (1-\beta_k)\mathbf{n}_j$, and $\mathbf{n}_j$ is a randomly chosen negative feature from $\tilde{Q}^N$, while $\beta_k \in (0, 0.5)$ is a randomly chosen mixing coefficient for the key. Ablations showed that this yields at best performance as good as mixing with the query, but on average about 0.1-0.2% lower.

**Weighted contributions for the logits of $\mathbf{h}'$.** We also tried weighing the contributions of the MoCHi samples according to the percentage of the query they have. That is, the logits of each hard negative $\mathbf{h}'_k$ was scaled by $\beta_k$ to reflect how "negative" this point is. This weighing scheme also resulted in slightly inferior results.

**Sampling negatives non-uniformly.** We also experimented when sampling negatives with a probability defined over a function of the logit values. That is, we defined a probability function by adding a softmax on top of the top $N$ negatives, with a $\tau'$ hyper-parameter. Although we would want to further investigate and thoroughly ablate this approach, early experiments showed that the

hard negatives created this way are so hard that linear classification performance decreases by a lot (10-30% for different values of $\tau'$).

**Fixing the *percentage* of hard negatives in the top-k logits.** In an effort to reduce our hyperparameters, we run preliminary experiments on a variant where instead of $s$ and $s'$, we synthesize hard negatives sequentially for each query by alternate between the two mixing methods (*i.e.* mixing two negatives and mixing the query with one negative) up until $X\%$ of the top-$N$ logits correspond to synthetic features. Although encouraging, quantitative results on linear classification were inconclusive; we would however want to further investigate this in the future jointly with curriculum strategies that would decrease this percentage over time.

### B.3 Mixing hard negatives vs altering the temperature of the softmax

Another way of making the contrastive loss more or less "peaky" is through the temperature parameter $\tau$ of the softmax function; we see from Eq (2) that the gradients of the loss are scaled by $1/\tau$. One would therefore assume that tuning this parameter could effectively tune the hardness and speed of learning. One can see MoCHi as a way of going beyond one generic temperature parameter; we start with the best performing, cross-validated $\tau$ and generate different negatives *adapted* to each query, and therefore have adaptive learning for each query that further evolves at each epoch.

## Appendix C  More experiments and results

### C.1 Experimental protocol

In general and unless otherwise stated, we use the default hyperparameters from the official implementation[2] for MoCo-v2. We follow Chen et al. [10] and use a cosine learning rate schedule during self-supervised pre-training. For both pretraining datasets the initial learning rate is set to 0.03, while the batchsize is 128 for ImageNet-100 and 256 for ImageNet-1K. Similar to MoCo-v2 we keep the embedding space dimension to 128. We train on 4 GPU servers. We further want to note here that, because of the computational cost of self-supervised pre-training, 100 epoch pretraining results are computed from the 100th-epoch checkpoints of a 200 epoch run, *i.e.* the cosine learning rate schedule still follows a 200 epoch training. Moreover, our longer (800) epoch runs are by restarting training from the 200 epoch run checkpoint, and switching the total number of epochs to 800, *i.e.*, the learning rate jumps back up after epoch 200.

#### C.1.1  Dataset details

**Imagenet.** The ImageNet-1K data can be downloaded from this link[3] while the 100 synsets/classes of ImageNet-100 are presented below. For ImageNet-1K the training set is 1.2M images from 1000 categories, while the validation set contains 50 images from each class, *i.e.* 50,000 images in total.

**ImageNet-100.** ImageNet-100 is a subset of ImageNet-1K that consists of the 100 classes presented right below. It was first used in Tian et al. [46] and recently also used in Shen et al. [40]. The synsets of ImageNet-100 are:

n02869837 n01749939 n02488291 n02107142 n13037406 n02091831 n04517823 n04589890
n03062245 n01773797 n01735189 n07831146 n07753275 n03085013 n04485082 n02105505
n01983481 n02788148 n03530642 n04435653 n02086910 n02859443 n13040303 n03594734
n02085620 n02099849 n01558993 n04493381 n02109047 n04111531 n02877765 n04429376
n02009229 n01978455 n02106550 n01820546 n01692333 n07714571 n02974003 n02114855
n03785016 n03764736 n03775546 n02087046 n07836838 n04099969 n04592741 n03891251
n02701002 n03379051 n02259212 n07715103 n03947888 n04026417 n02326432 n03637318
n01980166 n02113799 n02086240 n03903868 n02483362 n04127249 n02089973 n03017168
n02093428 n02804414 n02396427 n04418357 n02172182 n01729322 n02113978 n03787032
n02089867 n02119022 n03777754 n04238763 n02231487 n03032252 n02138441 n02104029
n03837869 n03494278 n04136333 n03794056 n03492542 n02018207 n04067472 n03930630

n03584829 n02123045 n04229816 n02100583 n03642806 n04336792 n03259280 n02116738
n02108089 n03424325 n01855672 n02090622

**PASCAL VOC.** For the experiments on PASCAL VOC we use the setup and config files described in MoCo's `detectron2` code[4]. The PASCAL VOC dataset can be downloaded from this link[5]. As mentioned in the main text, we fine-tune a Faster R-CNN [39], R50-C4 on `trainval07+12` and test on `test2007`. Details on the splits can be found here[6] for the 2007 part and here[7] for the 2012 part.

**COCO.** We similar to PASCAL VOC, we build on top of MoCo's `detectron2` code. We fine-tune all layers end-to-end on the `train2017` set (118k images) and evaluate on `val2017`. The image scale is in [640, 800] pixels during training and is 800 at inference.

### C.2 More ablations and results on ImageNet-100

Table 1 presents a superset of the ablations presented in the main text. Please note that most of the results here are *for a single run*. Only results that explicitly present standard deviation were averaged over multiple runs. Some further observations from the extended ablations table:

- From the 100 epoch run results, we see that the gains over MoCo-v2 get larger with longer training. When training longer (see line for 800 epoch pre-training), we see that MoCHi keeps getting a lot stronger, and actually seems to really close the gap even to the supervised case.
- Looking at the smaller queue ablation, we see that MoCHi can achieve with K=4k performance equal to MoCo-v2 with K=16k.
- From the runs using "class oracle" (bottom section), *i.e.* when simply discarding false negatives from the queue, we see that MoCHi comes really close to the supervised case, showing the power of contrastive learning with hard negatives also when labels are present.

### C.3 Results for ImageNet-1K

In Table 2 we present a superset of the results presented in the main text, for linear classification on ImageNet-1K and PASCAL VOC. We see that, for 200 epoch training performance still remains strong even when $N = 64$, while the same stands for $N = 128$ when training for 100 epochs. We also added a couple of recent and concurrent methods in the table, *e.g.* PCL [28], or the clustering approach of [7]. Both unfortunately use a different setup for PASCAL VOC and their VOC results are not directly comparable. We see however that our performance for linear classification on ImageNet-1K is higher, despite the fact that both methods take into account the class label-based *clusters* that do exist in ImageNet-1K.

**Oracle run for MoCHi.** We also present here a (single) run for MoCHi with a class oracle, when training on ImageNet-1K for 1000 epochs. From this very preliminary result we verify that discarding false negatives leads to significantly higher linear classification performance for ImageNet-1K, the training dataset, while at the same time state-of-the-art transfer learning performance on PASCAL VOC is preserved.

## Appendix D  An extended related and concurrent works section

Although self-supervised learning has been gaining traction for a few years, 2020 is undoubtedly the year when the number of self-supervised learning papers and pre-prints practically exploded. Due to space constraints, it is hard to properly reference all recent related works in the area in Section 2 of the main text. What is more, a large number of concurrent works on contrastive self-supervised learning came out after the first submission of this paper. We therefore present here an extended

related work section that complements that Section (works mentioned and discussed there are not copied also here), a section that further catalogues a large number of concurrent works.

Following the success of contrastive self-supervised learning, a number of more theoretical studies have very recently emerged, trying to understand the underlying mechanism that make it work so well [50, 53, 41, 27, 48], while Mutual Information theory has been the basis and inspiration for a number such studies and self-supervised learning algorithms during the last years, *e.g.* [21, 51, 56, 4, 13, 20]. Building on top of SimCLR [9], the Relational Reasoning approach [35] adds a reasoning module after forming positive and negative pairs; the features for each pair are concatenated and passed through an MLP that predicts a relation score. In RoCL [23], the authors question the need for class labels for adversarially robust training, and present a self-supervised contrastive learning framework that significantly improved robustness. Building on ideas from [30] where objectives are optimized *locally*, in LoCo [59] the authors propose to locally train overlapping blocks, effectively closing the performance gap between local learning and end-to-end contrastive learning algorithms.

**Self-supervised learning based on sequential and multimodal visual data.** A number of earlier works that learn representations from videos utilized the sequential nature of the temporal dimension, *e.g.* future frame prediction and reconstruction [42], shuffling and then predicting or verifying the order of frames or clips [33, 26, 60], predicting the "arrow of time" [55], pace [52] or predicting the "odd" element [14] from a set of clips. Recently, contrastive, memory-based self-supervised learning methods were extended to video representation learning [18, 20, 45, 49]. In an interesting recent study, Purushwalkam and Gupta [37] study the robustness of contrastive self-supervised learning methods like MoCo [19] and PIRL [32] and saw that despite the fact that they learn occlusion-invariant representations, they fail to capture viewpoint and category instance invariance. To remedy that, they present an approach that leverages unstructured videos and leads to higher viewpoint invariance and higher performance for downstream tasks. Another noteworthy paper that learns visual representations in a self-supervised way from video is the work of Emin Orhan *et al*. [34], that utilized an egocentric video dataset recorded from the perspective of several young children and demonstrated the emergence of high-level semantic information.

A number of works exploit the audio-visual nature of video [25, 1, 36, 11, 2] to learn visual representation, *e.g.* via learning intra-modality synchronization. Apart from audio, other methods have used use automatic extracted text, *e.g.* from speech transcripts [44, 43, 31] or surrounding text [16].

**Clustering losses.** A number of recent works explore representation learning together with clustering losses imposed on the unlabeled dataset they learn on. Although some care about the actual clustering performance on the training dataset [15, 38, 66], others further use the clustering losses as means for learning representations that generalize [6–8, 63, 62, 3]. Following the recent success of contrastive learning approaches, very recently a number of methods try to get the best of both worlds by combining contrastive and clustering losses. Methods like local aggregation [67], Prototypical Contrastive learning [28], Deep Robust Clustering [66], or SwAV [8] are able to not only create transferable representations, but also are able to reach linear classification accuracy on the training set that is not very far from the supervised case. In a very recent work, Wei *et al*. [54] introduce a consistency regularization method on top of instance discrimination, where they encourage the the similarities of the query and the negatives to match those of the key and the negatives, *i.e.* treating them as a soft distribution over pseudo-labels.

**Focusing on the positive pair.** Works like SimCLR [9], MoCo-v2 [10] and Tian *et al*. [47] make it clear that for contrastive self-supervised learning, selecting a challenging and diverse set of image transformations can highly boost the quality of representations. Recently, papers like SwAV [8, 61] demonstrated that even higher gains can be achieved by using multiple augmentations. In a very interesting concurrent work, Cai et al. [5] propose a framework where key generation is probabilistic and enables learning with infinite positive pairs in theory, while showing strong quantitative gains.

Very recently the BYOL [17] showed that one can learn transferable visual representations via bootstrapping representations and *without* negative pairs. Reproducibility studies[8] as well as very recent manuscripts [48] have shown that batch normalization might play an important role when learning without negatives, preventing mode collapse and helping spread the resulting features in the

embedding space. BYOL makes a number of modifications over SimCLR [9], *e.g.* the addition of a target network whose parameter update is lagging similar to MoCo [19] and a predictor MLP. They further use a different optimizer (LARS) and overall report transfer learning results after 1000 epochs with a batchsize of 4096, a setup that is almost impossible to reproduce (the authors claim training takes about 8 hours on 512 Cloud TPU v3 cores). It is hard to directly compare MoCHi to BYOL, as BYOL does not report transfer learning results for the commonly used setup, *i.e.* after 200 epochs of pre-training. We argue that by employing hard negatives, MoCHi can learn strong transferable representations faster than BYOL.

**Synthesizing for supervised metric learning.**    Recently, synthesizing negatives was explored in metric learning literature [12, 65, 24]. Works like [12, 65] use generators to synthesize negatives in a supervised scenario over common metric learning losses. Apart from not requiring labels, in our case we focus on a specific contrastive loss and exploit its *memory* component. What is more, and unlike [12, 65], we do not require a generator, *i.e.* have no extra parameters or loss terms that need to be optimized. We discuss the relations and differences between MoCHi and the closely related Embedding Expansion [24] method in the related works Section of the main paper.

**The MoCHi oracle and supervised contrastive learning.**    A number of recent approaches have explored the connections between supervised and contrastive learning [22, 64, 29]. Very recently, Khosla et al. [22] show that training a contrastive loss in a supervised way can lead to improvements even over the ubiquitous cross-entropy loss. Although definitely not the focus and merely a byproduct of the class oracle analysis of this paper, we also show here that MoCo and MoCHi can successfully perform supervised learning for classification, by simply discarding same-class negatives. This is something that is further utilized in [64]. For MoCHi, we can further ensure that all *hard negatives* come from other classes. In the bottom section of Table 1 we see that for ImageNet-100, the gap between the cross-entropy and contrastive losses closes more and more with longer contrastive training with harder negatives. An oracle run is also shown in Table 2 for ImageNet-1K after 1000 epochs of training. We see that MoCHi decreases the performance gap to the supervised for linear classification on ImageNet-1K, and performs much better than the supervised pre-training model for object detection on PASCAL. We want to note here that MoCHi oracle experiments on ImageNet-1K are very preliminary, and we leave further explorations on supervised contrastive learning with MoCHi as future work.

## Footnotes

[1] https://github.com/SsnL/align_uniform/

[2]https://github.com/facebookresearch/moco/

[3]http://image-net.org/challenges/LSVRC/2011/

[4]https://github.com/facebookresearch/moco/tree/master/detection

[5]http://host.robots.ox.ac.uk/pascal/VOC/

[6]http://host.robots.ox.ac.uk/pascal/VOC/voc2007/dbstats.html

[7]http://host.robots.ox.ac.uk/pascal/VOC/voc2012/dbstats.html

[8]https://untitled-ai.github.io/understanding-self-supervised-contrastive-learning.html

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

| Method | Top1 % ($\pm\sigma$) | diff (%) |
|---|---|---|
| *100 epoch training* | | |
| MoCo-v2 [10]* | 73.0 | |
| + MoCHi (1024, 1024, 128) | 73.6 | 0.6 |
| + MoCHi (1024, 128, 256) | 73.7 | **0.7** |
| *200 epoch training* | | |
| MoCo [19] | 73.4 | |
| MoCo + iMix [40] | 74.2$^{\ddagger}$ | 0.8 |
| CMC [46] | 75.7 | |
| CMC + iMix [40] | 75.9$^{\ddagger}$ | 0.2 |
| MoCo [19]* ($t = 0.07$) | 74.0 | |
| MoCo [19]* ($t = 0.2$) | 75.9 | |
| MoCo-v2 [10]* | 78.0 ($\pm0.2$) | |
| + MoCHi (16384, 1024, 0) | 78.1 | 0.1 |
| + MoCHi (16384, 0, 1024) | 78.5 | 0.4 |
| + MoCHi (16384, 0, 256) | 78.7 | 0.7 |
| + MoCHi (2048, 1024, 0) | 78.2 | 0.2 |
| + MoCHi (2048, 512, 0) | 78.5 | 0.5 |
| + MoCHi (2048, 512, 256) | 78.4 | 0.4 |
| + MoCHi (1024, 1024, 0) | 78.8 | 0.8 |
| + MoCHi (1024, 1024, 128) | **79.0** ($\pm0.4$) | **1.0** |
| + MoCHi (1024, 512, 0) | 78.9 | 0.9 |
| + MoCHi (1024, 0, 512) | 79.0 | 1.0 |
| + MoCHi (1024, 256, 512) | **79.0** ($\pm0.4$) | **1.0** |
| + MoCHi (1024, 128, 256) | **78.9** ($\pm0.5$) | **0.9** |
| + MoCHi (1024, 128, 0) | 78.8 | 0.8 |
| + MoCHi (1024, 0, 128) | 78.7 | 0.7 |
| + MoCHi (1024, 0, 256) | 78.9 | 0.9 |
| + MoCHi (1024, 0, 512) | 79.0 | 1.0 |
| + MoCHi (512, 128, 0) | 78.4 | 0.4 |
| + MoCHi (512, 512, 0) | 78.2 | 0.2 |
| + MoCHi (512, 128, 128) | 78.6 | 0.6 |
| + MoCHi (256, 256, 0) | 78.1 | 0.1 |
| + MoCHi (256, 512, 0) | 77.7 | 0.3 |
| + MoCHi (128, 128, 0) | 77.7 | 0.3 |
| + MoCHi (64, 6 4, 0) | 77.5 | 0.5 |
| $K = 4096$ | | |
| MoCo-v2 [10]* | 77.5 | |
| + MoCHi (1024, 1024, 128) | 78.0 | 0.5 |
| $K = 1024$ | | |
| MoCo-v2 [10]* | 76.0 | |
| + MoCHi (1024, 1024, 128) | 77.0 | 1.0 |
| + MoCHi (1024, 1024, 128) (all queue) | 73.4 | 2.6 |
| *800 epoch training* | | |
| MoCo-v2 [10]* | 84.1 | |
| MoCo-v2 [10] + MoCHi (1024, 1024, 128) | **84.5** | |
| Using Class Oracle | | |
| MoCo-v2* (200 epochs) | 81.8 | |
| + MoCHi (1024, 1024, 128) (200 epochs) | 82.5 | |
| + MoCHi (1024, 1024, 128) (400 epochs) | 84.2 | |
| + MoCHi (1024, 1024, 128) (800 epochs) | 85.2 | |
| Cross-entropy classification (supervised) | 86.2 | |

Table 1: More MoCHi ablations on ImageNet-100. Rows without a citation denote reproduced results. $^{\ddagger}$ denote results from Figure 4 in [40]. Unless standard deviation is explicitly reported, results in this table are for a single run.

| Method | IN-1k Top1 | VOC 2007 AP$_{50}$ | AP | AP$_{75}$ |
|---|---|---|---|---|
| *100 epoch training* | | | | |
| MoCo-v2 [10]* | 63.6 | 80.8 (±0.2) | 53.7 (±0.2) | 59.1 (±0.3) |
| + MoCHi (512, 1024, 512) | 63.7 | **81.3** (±0.1) **(0.6)** | **54.7** (±0.4) **(1.1)** | **60.6** (±0.5) **(1.6)** |
| + MoCHi (256, 512, 0) | 63.9 | 81.1 (±0.1) (0.4) | 54.3 (±0.3) (0.7) | 60.2 (±0.1) (1.2) |
| + MoCHi (256, 512, 256) | 63.7 | **81.3** (±0.1) **(0.6)** | **54.6** (±0.3) **(1.0)** | **60.7** (±0.8) **(1.7)** |
| + MoCHi (128, 1024, 512) | 63.4 | 81.1 (±0.1) (0.4) | **54.7** (±0.3) **(1.1)** | **60.9** (±0.1) **(1.9)** |
| *200 epoch training* | | | | |
| SimCLR [9] (8k batch size, from [10]) | 66.6 | | | |
| DeeperCluster [7] ([‡‡]train only on VOC 2007) | 48.4 | 71.9[‡‡] | | |
| MoCo + Image Mixture [40] | 60.8 | 76.4 | | |
| InstDis [57][†] | 59.5 | 80.9 | 55.2 | 61.2 |
| MoCo [19] | 60.6 | 81.5 | 55.9 | 62.6 |
| SeLa [3] | 61.5 | | | |
| PIRL [32][†] | 61.7 | 81.0 | 55.5 | 61.3 |
| InterCLR [58] | 65.5 | | | |
| PCL [28] ([‡]frozen body) | 65.9 | 78.5[‡] | | |
| PCL v2 [28] | 67.6 | | | |
| MoCo-v2 [10] | 67.7 | 82.4 | 57.0 | 63.6 |
| MoCo-v2 + CO2 [54] | 68.0 | 82.7 | 57.2 | 64.1 |
| InfoMin Aug. [47] | 70.1 | 82.7 | **57.6** | **64.6** |
| MoCo-v2 [10]* | 67.9 | 82.5 (±0.2) | 56.8 (±0.1) | 63.3 (±0.4) |
| + MoCHi (1024, 256, 128) | 68.0 | 82.3 (±0.2) (0.2) | 56.8 (±0.1) ( 0.0) | 63.8 (±0.4) (0.5) |
| + MoCHi (1024, 512, 256) | 68.0 | 82.3 (±0.2) (0.2) | 56.7 (±0.2) (0.1) | 63.8 (±0.2) (0.5) |
| + MoCHi (1024, 0, 512) | 67.8 | 82.7 (±0.1) (0.2) | 57.0 (±0.1) (0.2) | 64.0 (±0.2) (0.7) |
| + MoCHi (512, 1024, 512) | 67.6 | 82.7 (±0.1) (0.2) | 57.1 (±0.1) (0.3) | 64.1 (±0.3) (0.8) |
| + MoCHi (256, 512, 0) | 67.7 | **82.8** (±0.2) (0.3) | 57.3 (±0.2) (0.5) | 64.1 (±0.1) (0.8) |
| + MoCHi (256, 512, 256) | 67.6 | 82.6 (±0.2) (0.1) | 57.2 (±0.3) (0.4) | 64.2 (±0.5) (0.9) |
| + MoCHi (256, 2048, 2048) | *67.0* | 82.5 (±0.1) ( 0.0) | 57.1 (±0.2) (0.3) | 64.4 (±0.2) (1.1) |
| + MoCHi (128, 1024, 512) | *66.9* | 82.7 (±0.2) (0.2) | 57.5 (±0.3) (0.7) | 64.4 (±0.4) (1.1) |
| + MoCHi (64, 1024, 512) | *66.3* | 82.6 (±0.1) (0.1) | 57.3 (±0.1) (0.5) | 64.4 (±0.5) (1.1) |
| *800 epoch training* | | | | |
| SvAV [8] | *75.3* | 82.6 | 56.1 | 62.7 |
| MoCo-v2 [10] | 71.1 | 82.5 | **57.4** | 64.0 |
| MoCo-v2[10]* | 69.0 | 82.7 (±0.1) | 56.8 (±0.2) | 63.9 (±0.7) |
| + MoCHi (128, 1024, 512) | 68.7 | **83.3** (±0.1) (0.6) | 57.3 (±0.2) (0.5) | **64.2** (±0.4) (0.3) |
| *Using Class Oracle* | | | | |
| Cross-entropy classification (supervised) | 76.1 | 81.3 | 53.5 | 58.8 |
| MoCo-v2 [10] + MoCHi (512, 1024, 512) | 72.6 | *83.3* | *57.7* | *64.6* |

Table 2: Results on ImageNet-1K and PASCAL VOC. Rows that do not report standard deviation correspond to single runs. Wherever standard deviation is reported for the VOC fine-tuning, results are averaged over three runs. For MoCHi runs we also report difference to MoCo-v2 in parenthesis. * denotes reproduced results. [†] results are copied from [19].