[Reviews · NeurIPS 2020]

Review 1

Summary and Contributions: This paper does something similar to mixup (mixing two samples in order to generate a new one) in the context of MoCo. It mixes the hardest negatives in the MoCo queue in order to surround the query and force cleaner separation. It further mixes the hard negatives with the query itself, showing experimentally this makes the task even harder. This technique offers modest, but consistent, improvements.

Strengths: - The method seems to offer a consistent improvement over MoCo, with minimal extra computational costs (and made up for with better and faster learning). - Using the oracle gives an interesting sample point. - Varied experiments. I also appreciate the effort to have error bars, and I acknowledge that it is always a resource problem. - Good to consider the broader impact. Although the hope that this technique can contribute to reduced dataset bias is highly speculative and I wouldn't want to promote a trend where every algorithm starts to claims this, drowning out important work that actually tackles this problem and demonstrates it experimentally.

Weaknesses: - It would be great if you could try to give the reader more intuition about the mixing with the query technique. It just seems that it would simply rescale the loss in a certain way. If you actually plug the new h into (1) and expand it, the squared L2 norm of q shows up. However, the L2 norm of q should be 1, since the paper states that "all embeddings are L2-normalized". So, instead of "q^Tn" we get something like "beta + (1 - beta)q^Tn". The beta can be go out of the sum. If you follow this thought, I think you can see another way of formulating this that boils down to a change in the shape of the loss, or perhaps a "2-parameter temperature". I list this as a weakness, because I think this should be understood and communicated to the reader a bit more in depth, since it will facilitate future work. - The paper keeps saying this is faster, but maybe I'm missing exactly where this is thoroughly demonstrated. In Fig(a) it does show a bigger gap after 100 epochs than 200 epochs, which suggests this. However, a few more sample points and a plot that gives a sense of shape of the curves would strengthen this argument. MoCHi may give better results faster, but does it also reach a plateau earlier than MoCo-v2? If not, many people would still want to train for the same amount, which wouldn't lead to faster training in practice (better results though).

Correctness: Yes, the method is correct. I wonder if the mixing with the query technique can be reformulated in a more intuitive way though, as mentioned under weaknesses.

Clarity: Overall, yes. Although figures could be made more stand-alone from the text and a bit easier to interpret. More information on this in a separate field.

Relation to Prior Work: Yes, and I think it was good that mixup was mentioned.

Reproducibility: Yes

Additional Feedback: - Figure 2 (b/c) were hard to parse. Some descriptions are only in the text and not the caption or the legend. The dashed lines with triangles do not include synthetic features, so isn't that simply MoCo-v2 then? Since they are not included, the green and red MoCHi should be identical, right? So this is just two runs of the baseline, which is why they are so close? I'm also confused by the notion that the green line is faster - Fig 3c: The diff is not '%' but 'percentage points'.


Review 2

Summary and Contributions: This paper focuses on hard negative mining (or more accurately mixing) for self-supervised contrastive learning. Unlike finding hard negative samples, the authors create hard negatives by mixing pre-computed features, which does not require significant computational overhead. == update === I appreciate the results on MS COCO, and I also think it's more important to see improvements on transfer learning tasks than on ImageNet linear evaluation. According to my experience, such improvement shown in rebuttal on MS COCO is non-trivial, thus I upgrade my score from 6 to 7.

Strengths: The way to create hard negatives proposed in this paper is mixing hard negatives at the feature level, for each query point. This relates to the Manifold Mixup paper [29], which originally targets for supervised learning. So hard negative mining, as well as the way the authors proposed, is some new contribution to contrastive learning. (Concerns will be explained later) The empirical analysis of the evolution of hard negative samples of contrastive learning is actually interesting, and it may further inspire the way future researchers look at hard negative mining.

Weaknesses: In general, my biggest concern is that the efficacy of the proposed method is not very significant. - For example, it only improves over MoCo-v2 on ImageNet-100 with 0.8% and on ImageNet-1k with 0.1 (67.9% v.s. 68.0). - And the improvement on PASCAL VOC detection is also not very significant. - I also wonder whether the proposed approach is compatible with more advanced data augmentation that sets a higher baseline. For example, can the proposed module directly be dropped in to improve [25] and [28]? In line 233-234, the explanation here is a bit counter-intuitive. I understand this is possible if inter-class distance increases more than intra-class variance, but can you plot it to justify it better? Otherwise, increasing intra-class variance should lead to lower accuracy. As shown in Figure 3(b), why are s' and s not comparable. For example, the last column shows that increase s' actually harms? It would be great to have deeper understanding on how s and s' are really manipulating the latent space (or logits). Why in section 4, authors always argues for light weight computation, I wonder how big is the difference of running time (e.g. relatively x% slower)? Because ranking the logits and mix features from noncontinuous GPU ram (as the top-N negatives are not continuous in the queue) should take some time. The line 141-150 need to be revised. Indeed, the pre-text task accuracy drop between MoCo and MoCo v2, as shown in Figure 2(b), mostly results from non-linear projection head rather than data augmentation, though augmentation also contributes a bit. Correct me if I am wrong. How can p goes beyond 1 in Figure 2(a)?

Correctness: Looks reasonable to me.

Clarity: Not very well written. It really slows down my reviewing speed significantly. I found myself need to read back and forth multiple times to understand the organization better. Generally the writing should be improved. Figure 2 is messy. - (a) the probability should not go beyond 1?? - legend in (b) missing some items, which confuses me when I read the text describing it - The color and shape in (c) is also less clear. Figure 3: - (a) maybe label X-axis as N to be clearer? = (b) maybe label s and s' in the table? Figure 4: - Caption: "results copied from [13]". I did not find these results from [13], from somewhere else?

Relation to Prior Work: The authors are up-to-date, and includes even recent related works. So prior works are clearly discussed and the difference is clear.

Reproducibility: Yes

Additional Feedback: Maybe also includes MS-COCO results? I am giving marginally above threshold, as I think the negative analysis is interesting, but the efficacy of the proposed mixing method is less effective, unfortunately.


Review 3

Summary and Contributions: This paper considers the problem of self-supervised learning. Specifically, the authors construct their model based on MOCO and argue that hard negative samples are important in improving performance. Therefore, the authors present a strategy to generate virtual hard negative samples by mixing among top-ranked samples. Experiments show the effectiveness of the proposed method in self-supervised learning setting.

Strengths: 1. This paper considers a specif problem in MOCO where the authors aim to generate new hard negative samples for learning more discriminative features. The proposed method is simple but can consistently improve the results. 2. The authors provide a good analysis to show the importance of hard negative samples in Sec 3.2 and 3.3. This helps readers to better understand the motivation of this paper. 3. Throughout experiments are provided to demonstrate the effectiveness of the proposed method. Consistent improvements are obtained by the proposed mixing methods.

Weaknesses: 1. Although this is the first (If I am right) work that considers the hard negative samples in self-supervised learning, the proposed method is somewhat simple and trivial. The main framework is the MOCO and the proposed hard negative generating is based on the well-known mixup. This is my major concern. 2. From Fig. 3b, I found that the proposed two mixing strategies are not well-complementary to each other. Using only one can achieve the highest results. Therefore, I I don’t think it is necessary to jointly using these two strategies. 3. From Fig.3 c, the iMix [25] can also improve the performance of MOCO. How about the reuslts when applying iMIX on MOCO V2. Can it achieve similar improvement to the proposed MoCHI? In addition, I think the proposed MoCHI is very similar to iMIX[25]. This paper does not introduce too much new techniques compared to iMIX.

Correctness: Yes. It is simple and clearly correct.

Clarity: Yes. This paper is well written and easy to follow.

Relation to Prior Work: Yes. This paper provides its own motivation and discuss the difference to other works. One mix-based method could be discussed in the related work [A]. OpenMix: Reviving Known Knowledge for Discovering Novel Visual Categories in An Open World. Arxiv 2020.

Reproducibility: Yes

Additional Feedback: Post rebuttal: I have read the comments of other reviewers and the rebuttal. The authors have addressed my concerns. Although the proposed method is simple, the authors provide good intuition and analyst for why negative samples matter in self-supervised learning. The proposed method achieves consistent improvement in downstream tasks. I also agree with R2 that the improvement in downstream tasks is important. To this end, I would like to upgrade my rating to 6.


Review 4

Summary and Contributions: This paper proposes to generate synthetic hard negatives for contrastive learning by mixing the real hard negatives. The authors also provide an in-depth analysis about hard negatives sampling in contrastive learning tasks, and justify why sampling/generating harder negatives is needed. The experiments follow standard self-supervised learning benchmarks and implementations, however the relative accuracy improvement is not very high.

Strengths: Simple but interesting method to generate more synthetic hard negatives for a given set of anchor and negative points. The authors also provide a detailed analysis with the oracle in order to justify the need for their method. The paper is well written and is easy to understand.

Weaknesses: The major weakness of the paper is the relative improvement in terms of accuracy in the experiments. The proposed method only improves the baseline by 1% accuracy in ImageNet-100, and less than 1% in ImageNet-1K. The paper also misses similar existing work in metric learning. [1,2] (below) generate hard negatives in the context of supervised metric learning. [2] especially presents a similar solution by mixing the embeddings of the anchor and negatives. [1] Duan et al. Deep adversarial metric learning. CVPR 2018 [2] Zheng et al. Hardness-aware deep metric learning. CVPR 2019 After rebuttal: I appreciate the additional experiments on the MS COCO dataset where the improvement are more significant. I'm improving my rating from 5 to 6 after the rebuttal.

Correctness: Yes

Clarity: Yes

Relation to Prior Work: Discussion about some prior work is missing. Please see the Weaknesses.

Reproducibility: Yes

Additional Feedback:

[Author Response · NeurIPS 2020]

1. We want to thank all reviewers for their constructive feedback. Here, we reply to all questions and present more results
2. (on **COCO** in Tab. 1 and **after 800 epochs** in Tab. 2) that further support the efficacy of the proposed approach.
3. **Summary of contributions.** Despite the multitude of SSL papers during the months before *and after* the NeurIPS
4. submission deadline, we find that our contributions still stand, and we are glad that reviewers also generally agree
5. that: a) **hard negatives in contrastive SSL** are under-explored beyond MoCHi and our **empirical analysis** and oracle
6. experiments are interesting (**R1**, **R2**, **R3**) and may inspire future research (**R2**); b) **mixing for hard negatives** is novel
7. (**R1**, **R2**, **R3**) and produces **consistent gains** (**R1**, **R3**) over the state-of-the-art (SoTA) MoCO-v2 on 3 datasets (COCO,
8. VOC, Im-100) and 3 tasks; c) We report results that **set a new SoTA** also for shorter (100 epoch) pre-training. We are
9. also glad that reviewers **R1**, **R3** and **R4** found our paper clearly written and easy to read.

Table 1: Object detection and instance segmentation results **on COCO with the** $\times 1$ **training schedule** (C4 backbone).

| Pre-train | Epochs | $AP^{bb}$ | $AP_{50}^{bb}$ | $AP_{75}^{bb}$ | $AP^{mk}$ | $AP_{50}^{mk}$ | $AP_{75}^{mk}$ |
|---|---|---|---|---|---|---|---|
| Supervised [13] | | 38.2 | 58.2 | 41.6 | 33.3 | 54.7 | 35.2 |
| MoCo [13] | 200 | 38.5 | 58.3 | 41.6 | 33.6 | 54.8 | 35.6 |
| MoCo (1B image train) [13] | 200 | 39.1 | 58.7 | 42.2 | 34.1 | 55.4 | 36.4 |
| InfoMin Aug. [28] | 200 | 39.0 | 58.5 | 42.0 | 34.1 | 55.2 | 36.3 |
| MoCo-v2 [6] | 200 | 39.0 $_{(\pm 0.1)}$ | 58.6 $_{(\pm 0.1)}$ | 41.9 $_{(\pm 0.3)}$ | 34.2 $_{(\pm 0.1)}$ | 55.4 $_{(\pm 0.1)}$ | 36.2 $_{(\pm 0.2)}$ |
| + MoCHi (256, 512, 0) | 200 | 39.2 $_{(\pm 0.1)}$ ($\uparrow$0.2) | 58.8 $_{(\pm 0.1)}$ ($\uparrow$0.2) | 42.4 $_{(\pm 0.2)}$ ($\uparrow$0.5) | 34.4 $_{(\pm 0.1)}$ ($\uparrow$0.2) | 55.6 $_{(\pm 0.1)}$ ($\uparrow$0.2) | 36.7 $_{(\pm 0.1)}$ ($\uparrow$0.5) |
| + MoCHi (128, 1024, 512) | 200 | 39.2 $_{(\pm 0.1)}$ ($\uparrow$0.2) | 58.9 $_{(\pm 0.2)}$ ($\uparrow$0.3) | 42.4 $_{(\pm 0.3)}$ ($\uparrow$0.5) | 34.3 $_{(\pm 0.1)}$ ($\uparrow$0.2) | 55.5 $_{(\pm 0.1)}$ ($\uparrow$0.1) | 36.6 $_{(\pm 0.1)}$ ($\uparrow$0.4) |
| + MoCHi (512, 1024, 512) | 200 | **39.4** $_{(\pm 0.1)}$ (**$\uparrow$0.4**) | **59.0** $_{(\pm 0.1)}$ (**$\uparrow$0.4**) | **42.7** $_{(\pm 0.1)}$ (**$\uparrow$0.8**) | **34.5** $_{(\pm 0.0)}$ (**$\uparrow$0.3**) | **55.7** $_{(\pm 0.2)}$ (**$\uparrow$0.3**) | **36.7** $_{(\pm 0.1)}$ (**$\uparrow$0.5**) |
| MoCo-v2 [6] | 100 | 37.0 $_{(\pm 0.1)}$ | 56.5 $_{(\pm 0.3)}$ | 39.8 $_{(\pm 0.1)}$ | 32.7 $_{(\pm 0.1)}$ | 53.3 $_{(\pm 0.2)}$ | 34.3 $_{(\pm 0.1)}$ |
| + MoCHi (256, 512, 0) | 100 | 37.5 $_{(\pm 0.1)}$ ($\uparrow$0.5) | 57.0 $_{(\pm 0.1)}$ ($\uparrow$0.5) | 40.5 $_{(\pm 0.2)}$ ($\uparrow$0.7) | 33.0 $_{(\pm 0.1)}$ ($\uparrow$0.3) | 53.9 $_{(\pm 0.2)}$ ($\uparrow$0.6) | 34.9 $_{(\pm 0.1)}$ ($\uparrow$0.6) |
| + MoCHi (128, 1024, 512) | 100 | **37.8** $_{(\pm 0.1)}$ (**$\uparrow$0.8**) | **57.2** $_{(\pm 0.0)}$ (**$\uparrow$0.7**) | **40.8** $_{(\pm 0.2)}$ (**$\uparrow$1.0**) | **33.2** $_{(\pm 0.0)}$ (**$\uparrow$0.5**) | **54.0** $_{(\pm 0.2)}$ (**$\uparrow$0.7**) | **35.4** $_{(\pm 0.1)}$ (**$\uparrow$1.1**) |

10. **Does MoCHi learn faster? (R1, R2)** Yes! To further justify the learning speed of MoCHi beyond the training curves
11. of Fig.2, we report results on VOC (Fig. 4a) and also COCO (Tab. 1) after 100 epoch pre-training, *i.e.* half of the
12. standard 200 epochs. MoCHi reachs *performance similar to supervised pre-training (33.2) in 100 epochs* for instance
13. segmentation on COCO. We further measured our computational overhead in terms of wall-clock time (**R2**) for runs
14. with/out MoCHi and found that MoCHi training was approx. 5-25% slower (different params/machines/loads).
15. **Are the gains significant? (R2, R3, R4)** To further support MoCHi's
16. efficacy, in Tab. 1 we report results on COCO (requested by **R2**), that
17. further show MoCHi achieving *consistent gains* over SoTA. We see

Table 2: Results after training for 800 epochs.

| Method | IN-1k Top1 | VOC 2007 $AP_{50}$ | AP | $AP_{75}$ |
|---|---|---|---|---|
| Supervised [13] | 76.1 | 81.3 | 53.5 | 58.8 |
| MoCo-v2 [6]* | 69.0 | 82.7 $_{(\pm 0.1)}$ | 56.8 $_{(\pm 0.2)}$ | 63.9 $_{(\pm 0.7)}$ |
| + MoCHi (128, 1024, 512) | 68.7 | **83.3** $_{(\pm 0.1)}$ | **57.3** $_{(\pm 0.2)}$ | **64.2** $_{(\pm 0.4)}$ |

18. that a) MoCHi outperforms the SoTA method of [28] by ~0.5%,
19. supervised pre-training by over 1%, and has higher gains over MoCo-
20. v2 when training for only 100 epochs. On VOC we see from Fig.4a
21. that a) gains are robust to different configurations b) gains are larger
22. (~1+%) for transferring after only 100 epoch training and c) from
23. Tab. 2 we see that *gains persist after longer training*; we report a new SoTA performance for transfer learning on VOC.
24. **Is MoCHi complementary to iMix [25] and [28]? (R2, R3)** We first want to respectfully remind the reviewers that
25. both are *unpublished*, concurrent works. [25] mixes *in image space* and does not deal with neither hardness nor the
26. negatives. Focusing on hard negatives allows MoCHi to a) show significant gains on transfer learning *after only 100*
27. *epochs*); b) achieve significantly higher performance than [25] for longer training (Tab. 1 in the supplementary and
28. Tab. 2). Our method performs on par with (and better than) [28] on VOC (and on COCO). We believe (but didn't have
29. time to verify) that better positive pairs (*e.g.* from [28]) are indeed complementary to MoCHi.
30. **Are the two mixing methods complementary? (R2, R3).** We agree that Fig. 3b gives mixed messages; this is because
31. it shows the *first* run per combination, not the mean (we didn't have multiple runs for all combinations, but only the top
32. ones). We will replace all numbers with means and variance (numbers we now have) and it will then be more clear,
33. together with the COCO and VOC results, that both are needed to get top performance.
34. **On missing works from metric learning (R4).** We thank the reviewer, these are indeed related works that we will
35. cite and discuss. The two paper mentioned by **R4**, similar to the even newer [15] from CVPR 2020, operate on metric
36. learning losses and in a supervised setting (see l97-101 for differences to [15]). We instead focus on the loss of Eq.(1),
37. have no class labels, and exploit the *memory*, something not present in metric learning works. What is more, both works
38. mentioned by **R4** require a generator, *i.e.* extra parameters and loss terms that need to be optimized; MoCHi has no
39. added parameters and further goes beyond triplets, by mixing negatives from the memory.
40. **On MoCHi being "simple and trivial" (R3).** We agree that MoCHi is a **simple** method; yet we argue that this is not
41. a weakness in itself, but a desirable characteristic given that it gives consistent gains. We also argue that it is **novel** in
42. the realm of SSL, as **R1**, **R2** and also **R3** themselves do. However, we respectfully disagree that it is **trivial**. We hope
43. we clarified the other two points listed as weaknesses by **R3** (novelty over the concurrent [25] and complementarity of
44. the mixing methods), but we cannot really rebut "trivial", a claim that can be neither properly justified, nor rebutted.
45. **Mixing the query as rescaling of the loss (R1).** Thanks, this is correct; we will add this intuition in the text.
46. **Other points by R2.** a) *Probability in Fig. 2(a) goes over 1*: There is a scaling factor of 1e-2 (top left corner); b) *Paper*
47. *organization/writing*: We agree that parts of the text can be more clearly presented and structured; we will reorganize
48. and revise all Figs; c) *L141-150*: a great point; we will run an experiment to directly evaluate and rephrase.

[Meta-Review · NeurIPS 2020]

All the reviewers agree this paper is strong. Due to the popularity of contrastive learning and the ideas in this paper, the paper will be timely at NeurIPS. The reviewers also praised the strong empirical results, especially over established baselines. The careful analysis and use of hard negative mining leads to these strong results, as noted by the reviewers.